**Data Availability Statement:** Data cannot be shared publicly because of GDPR, local protection act. This manuscript is based on health data. Access to these data is regulated by Finnish

# Intestinal parasites may be associated with later behavioral problems in internationally adopted children

**Anna-Riitta Heikkilä**[1]◉*, **Marko Elovainio**[2,3]◉, **Hanna Raaska**[4]◉, **Jaakko Matomäki**[5]◉, **Jari Sinkkonen**[6]◉, **Helena Lapinleimu**[7]◉

**1** Department of Pediatrics, University of Helsinki and Helsinki University Hospital, Helsinki, Finland, **2** Research Program Unit, Faculty of Medicine, University of Helsinki, Helsinki, Finland, **3** Finnish Institute for Health and Welfare, Helsinki, Finland, **4** Department of Child Psychiatry, Helsinki University Hospital, Helsinki, Finland, **5** Clinical Research Centre, Turku University Hospital, Turku, Finland, **6** Department of Child Psychiatry, University of Turku, Turku, Finland, **7** Department of Pediatrics and Adolescent Medicine, University of Turku and Turku University Hospital, Turku, Finland

◉ These authors contributed equally to this work.
* anna-riitta.heikkila@helsinki.fi

## Abstract

### Aim

At arrival in new home country, internationally adopted children often have intestinal parasites. International adoptees also exhibit more behavioral problems than their biological peers. We examined whether intestinal parasite infections in international adoptees on arrival in Finland are associated with their later behavioral and emotional problems.

### Methods

Data for this study were sourced from the Finnish Adoption Study (FinAdo) based on parental questionnaires for all internationally adopted children under 18 years ($n = 1450$) who arrived in Finland from 1985 to 2007. A total of 1293 families provided sufficient information on the adoptee's background, parasitic status on arrival, and behavioral symptoms at the median time of 5 years after arrival (mean age = 7.8 years). Behavioral and emotional disorders were evaluated with the Child Behavior Checklist (CBCL). Statistical analyses were conducted using linear regression.

### Results

Of the 1293 families, parents of 206 adoptive children reported intestinal parasites in their adopted children on arrival. Parasite-infected children had subsequently higher CBCL problem scores than the children without parasites ($p < 0.001$). The association between intestinal parasites and later behavioral problems was stronger than that between intestinal parasites and any other factors measured in this study, except disability.

### Limitations

The control group was naturally provided by the adopted children without parasite infections, but we could not compare the adopted children to non-adopted children without a defined

legislation and Findata, the Health and Social Data Permit Authority. The disclosure of data to third parties without explicit permission from Findata is prohibited. Only those fulfilling the requirements established by Finnish legislation and Findata for viewing confidential data are able to access the data. For further information and data access queries, please go to: https://www.findata.fi/en/about-us/data-protection-and-the-processing-of-personal-data/.

**Funding:** AH: The Foundation of Pediatric Research, Finland (personal grant, no specific grant number), https://www.lastentautientutkimussaatio.fi HL: Finland's Slot Machine Association (RAY) (personal grant, no specific grant number), https://www.stea.fi/web/en/frontpage; The Gyllenberg Association (personal grant, no specific grant number), https://gyllenbergs.fi; the Foundation for Pediatric Research, Finland (personal grant, no specific grant number), www.lastentautientutkimussaatio.fi; EVO Grant from Turku University Hospital (personal grant, no specific grant number), https://www.utu.fi/fi/tutkimus/tutkimusrahoituksen-tuki; The Tiukula foundation (personal grant, no specific grant number); the Yrjö Jahnsson Association (personal grant, no specific grant number) https://www.yjs.fi; Finnish Cultural Foundation, Varsinais-Suomi Regional fund (personal grant, no specific grant number) https://skr.fi/en received consultation fees from the following national adoption organizations in Finland: Interpedia, Save the Children Association Finland., and the international adoption service of the City of Helsinki. JS: The Tiukula Foundation (personal grant, no specific grant number); Finland's Slot Machine Association (RAY) (personal grant, no specific grant number), https://www.stea.fi/web/en/frontpage The funders had no role in study design, data collection and analysis, decision to publish, or preparation of the manuscript.

**Competing interests:** Helena Lapinleimu (MD, PhD) has received consultation fees from national adoption organizations in Finland (Interpedia, Save the Children Association Finland., and the international adoption service of the City of Helsinki). All the other authors have no conflicts of interest to declare.

parasite infection. We were unable to specify the effects associated with a specific parasite type. It was not possible either to include multiple environmental factors that could have been associated with behavioral problems in the models, which indicated only modest explanatory values.

## Conclusions

In this study, intestinal parasite infections in early childhood may be associated with children's later psychological wellbeing, even in children who move to a country with a low prevalence of parasites. Our findings may support further developments pertaining to the gut-brain theory.

## Introduction

Internationally adopted children often show growth and developmental delays on arrival in their new home country [1–6]. Many studies have also shown international adoptees to have more behavioral problems compared to their same-age, non-adopted peers [7,8]. Although many psychological explanations exist for growth and developmental delays also more physiological explanations are plausible. Intestinal parasite infections are common on arrival among internationally adopted children, especially among adoptees from countries with poor living conditions and subpar sanitation [3,4,6,9–16]. Intestinal parasites in children are associated with growth and developmental delays [17–19] as well as micronutrient deficiencies [20]. Micro-nutritional deficiencies, such as iron deficiency, are known to be associated with multiple developmental problems [21–23]. Developmental problems, in turn, have been shown to cause increased behavioral problems [24].

Intestinal parasites also affect the microbial imbalance of intestinal microbiota [25–27]. Intestinal parasites alter the intestinal physiology, cause chronic intestinal inflammation [28], and impact on the microbiota-gut-brain axis [27,29–31]. The gut microbiome affects brain function by changing immunological mechanisms, causing inflammation, and modifying the functions of several neurotransmitters [32–36]. Recent studies have also suggested that the intestinal microbiome plays a role in neurodevelopmental and neurobehavioral conditions [31,32,37], and some clues exist about their association with personality traits [33] and psychological symptoms [36].

No previous studies have shown whether the risks associated with earlier intestinal parasite infections persist after the child moves to more nurturing living conditions, when the intestinal parasitic infection has been efficiently treated, and when the child has had no reinfections. Therefore, in this study we investigated whether an association exist between intestinal parasites and later behavioral problems among internationally adopted children.

## Methods

### Participants

This work is part of the FINnish ADOption (FinAdo) study, which includes all children internationally adopted through legalized adoption organizations in Finland between 1985 and 2007. The survey data were collected through questionnaires between December 2007 and March 2009 and included two mailed questionnaires for non-respondents. The questionnaires covered child-related factors, family-related factors, background data of the adoptive parents

as well as the information of intestinal parasites of adopted children on arrival in Finland. As a part of the survey, the parents also completed the Child Behavior Checklist (CBCL) questionnaire [7,8] on the behavioral and emotional status of their children at the mean age of 7.8 (SD = 4.3) years, and the median age of 7 years. The median time after adoption in Finland was 5 years during the time the survey was conducted.

The original cohort included 1450 international adoptees (816 girls, 56%) under 18 years old (mean age = 7.5 years, SD = 4.4) at the commencement of study. The participation rate for the study was 55.7% [38]. The final sample comprises a total of 1293 under the age of 18 years (724 girls, 56%) who had been screened for the intestinal parasites and whose adoptive parents provided sufficient information about their behavioral characteristics during the evaluation (Fig 1).

The study was approved by the Ethics Review Committee of the Hospital District of Southwest Finland, and the participating children and their parents provided their written consent to participate.

## Measures

**Child-related factors.**   The child's gender, age at adoption, age at evaluation, and country of birth were reported by the parents. The type and number of placements before adoption were classified into three categories for statistical analysis. The place of origin was categorized by continent and region. Parents were also asked about any medical diagnoses their children received from Finnish medical evaluations soon after arrival or later.

Growth values of adopted children at arrival were reported in the parental questionnaire. For the analysis growth values were expressed as z-scores, which were compared against the growth charts of the World Health Organization [39]. Weight-for-age, height-for-age, and head circumference-for-age were used separately for both genders. The z-scores ranged from -6 SD to +3 SD for weight, from -6 SD to +2.5 SD for height, and from -5 SD to +3 SD for head circumference. Children with missing growth values were excluded from the fourth step of the linear regression analysis. z-scores below -2 SD in each category were considered to indicate delayed growth.

**Parasitic status.**   In Finland internationally adopted children undergo health and medical evaluations in public health care centers soon after arrival in the country. All the children were recommended tests for parasite infections by means of stool examinations for ova and

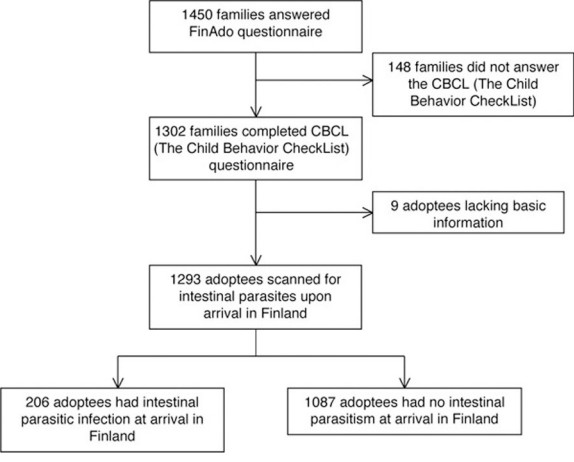

**Fig 1. Selection of the study population.**

parasites (2 specimens taken 24 to 48 hours apart) at the local health care centers. The parents were requested to obtain their children's test results from the health care centers and attach them with the completed questionnaire report. Thus, all the medical data were collected retrospectively from the parents via the questionnaire.

**Family-related factors.**   In addition, the parents answered a series of questions about their own marital and socioeconomic status and health. Each adoptive family's socioeconomic status was classified into one of four categories according to the vocation of the family member with the highest status. The marital status of each parent was classified into one of two categories.

**Behavioral problems.**   The CBCL was chosen to measure the adoptees' behavioral and emotional problems because of its wide use in adoption research [7,8], as well as the fact it has shown good psychometric reliability in studies on internationally adopted [8]. The CBCL provides a total score for behavioral problems and also separates the scores for internalizing and externalizing behavioral symptoms. Internalizing behavioral symptoms reflect problems mainly within the self, such as anxiety, depression, somatic complaints without medical cause, and withdrawal from social contacts. Externalizing behavioral signs include conflict with others, such as rule-breaking or aggressive behavior [40]. Each item was rated as 0 (not true), 1 (somewhat or sometimes true), and 2 (very true or often true). The higher the child's scores in the CBCL, the greater the extent of his/her behavioral problems.

In this study we used two CBCL questionnaires: the CBCL for ages 1.5 to 5 years with 100 questions and that for ages 6 to 18 years with 113 questions [8]. These questionnaires produce identical dimensions and total scores, thus providing comparable age-adjusted scores. Completed CBCL questionnaires were received for 1302 of the 1450 cases.

## Data analyses

Data obtained were analyzed using STATA program package (version 15) and R 3.6.1, and *p* values below 0.05 were considered statistically significant. All the significance tests were two-tailed. The child's current age, age on arrival, and CBCL scale were analyzed as continuous variables. Gender, continent of origin, placement before adoption, the child's disabilities, presence of anemia (Hb limit < 11.5 g/dL), socioeconomic status of the adoptive family, and the adoptive parent's marital status were used as the categorical variables.

Further analyses were done using linear regression in four steps. The first step tested for any association between the parasites and behavioral problems. In the second step, the association was adjusted for child-related factors. In the third step, the association was additionally adjusted for family-related factors. The final model also included information on anemia and the child's growth variables at adoption. For the sensitivity evaluation, the interacting effects between the child's intestinal parasite infection, age, gender, age at adoption, continent of birth, parental socioeconomic status, and parental marital status on the scales provided by the CBCL were assessed. The older the child at adoption, the longer the time spent in an orphanage or a foster home and the longer the exposure time to intestinal parasites. Therefore, the age of the child on arrival in Finland was used as a measure of exposure time in the analysis. Little is known about the pre-adoptive conditions of the children, but the early placement in an orphanage or foster home was considered in the models.

## Results

Of the 1293 children screened, 206 had at least one intestinal parasite on arrival in Finland (Fig 1), with *Giardia lamblia* being the most prevalent. Of the 206 children, 122 (59.2%) had *Giardia* in their stool samples, which collected after arrival, and 62 children (30.1%) had no

species listed at all. Of the children, 13 (6.3%) had *Entamoeba histolytica*, 7 (3.4%) had *Blastocystis hominis*, 5 (2.4%) had tapeworms, 4 (1.9%) had strongyloides, and 2 (1.0%) had *Hymenolepis nana*. Only 2 (1.0%) cases involved pinworms (*Enterobius vermicularis*), and 17 (8.3%) families omitted species of the worm found. Single cases emerged of *Ascaris lumbricoides*, Whipworm, or Schistosomiasis. Of the adoptees with intestinal parasitism, 7.8% had multiple intestinal parasites on arrival, but because of this number was low, the influence of multiple parasites was not analyzed separately.

The children with a detectable parasite infection at adoption were older (mean age = 2.6 years) than were those without parasites (mean age = 2.1 years; $p = 0.002$) (Table 1). At the time of the evaluation, the mean ages of the children with a parasite infection at arrival was 8.2 years, and for uninfected children, 7.7 years. Of the 206 adopted children with intestinal parasites on arrival, 51.9% were from Asia, 26.7% from Africa, 9.7% from South America, and 11.7% from Eastern Europe. The number and type of pre-adoption placements had no association with the prevalence of intestinal parasite infections (Table 2). In terms of the growth values, the presence of an intestinal parasite infection was associated only with sub-standard height ($p < 0.001$) (Table 1).

The adopted children with an intestinal parasite infection on arrival had significantly higher CBCL problem scores than their uninfected counterparts for all three categories: total problem scores ($p = 0.001$), externalizing problem scores ($p = 0.002$), and internalizing problem scores ($p = 0.003$). The associations found between the CBCL categories and parasites did not differ significantly between the parasite types, namely *Giardia* and helminth. Adjusting for child- or family-related factors did not affect the association. Among the intestinal parasite-infected adopted children, growth delays or anemia at adoption were not associated with higher CBCL problem scores for any child (Table 2). Significant interaction effects were observed between continents of birth and parasite infection for all CBCL categories: intestinal parasite infections increased external CBCL scores to a greater extent for children from South America ($p = 0.010$) than for children from Asia. For internalizing symptoms, intestinal parasite infections increased CBCL scores to a greater extent among children from South America ($p = 0.016$) and Eastern Europe ($p = 0.043$) than among those from Asia. The total CBCL symptoms scores were significantly higher ($p = 0.003$) among the parasite-infected adopted children from South America. The means of the externalizing symptoms, internalizing symptoms, and total CBCL symptoms were lower in parasite-infected children adopted from Asia than in those with a parasitic infection adopted from South America (9.6 vs 17.2, 6.2 vs 9.4, and 26.5 vs 45.3, respectively). Moreover, the mean of the internalizing symptoms (6.2) was lower in parasite-infected children adopted from Asia than in children with parasitic infection adopted from Eastern Europe (10.1).

## Discussion

In this study, we found that intestinal parasite infection upon entry to the country of adoption was associated with higher CBCL problem scores later in life (at the median time of 5 years after adoption). The reason behind this association is not known. Intestinal parasite infections may cause behavioral problems via multiple plausible biological mechanisms. Intestinal parasites may influence the maturing and developing brain by causing micronutrient deficiencies [20], or affecting gut-brain-axis by changing the intestinal microbiome [25–27], and causing intestinal inflammation [28]. Changes in the intestinal microbiota after antibiotic treatment [41,42] for intestinal parasite infections may also impact the children's brain. In any case, our results suggest that intestinal parasites are harmful in the early years of life to the internationally adopted children, who are undergoing rapid growth and development of the brain. These

Table 1. Characteristics of the FinAdo study sample presented as means (standard deviation) and numbers (percentage of the sample).

| Characteristic | All | Children with parasites | | Children with no parasites | | p value |
|---|---|---|---|---|---|---|
| | n (%)/Mean (SD) | n (%) | Mean (SD) | n (%) | Mean (SD) | |
| **Children** | 1293 (100) | 206 (15.9) | | 1087 (84.1) | | |
| Girls | 724 (56.0) | 103 (14.2) | | 621 (85.8) | | 0.059 |
| Boys | 569 (44.0) | 103 (18.1) | | 466 (81.9) | | |
| **Age on arrival in Finland, years** | 2.2 (1.9) | | 2.6 (1.6) | | 2.2 (1.9) | 0.002 |
| **Age at evaluation, years** | 7.8 (4.3) | | 8.2 (4.4) | | 7.7 (4.3) | 0.154 |
| **Disabilities*** | 137 (10.6) | 16 (11.7) | | 121 (88.3) | | 0.145 |
| **Anemia** | | | | | | 0.86 |
| No | 1174 (90.1) | 189 (16.1) | | 985 (83.9) | | |
| Yes | 110 (8.5) | 17 (15.5) | | 93 (84.5) | | |
| **Growth on arrival in Finland** | | | | | | |
| Weight (SD) | 1099 | 184 | -0.9 (1.2) | 915 | -0.8 (1.1) | 0.139 |
| Height (SD) | 993 | 165 | -1.8 (1.3) | 828 | -1.2 (1.3) | <0.001 |
| Head circumference (SD) | 765 | 108 | -0.4 (1.0) | 657 | -0.4 (1.1) | 0.923 |
| **Continent of birth** | | | | | | <0.001 |
| Asia | 712 (55.1) | 107 (15.0) | | 605 (85.0) | | |
| Africa | 141 (10.9) | 55 (39.0) | | 86 (61.0) | | |
| South America | 179 (13.8) | 20 (11.2) | | 159 (88.8) | | |
| Eastern Europe | 261 (20.2) | 24 (9.2) | | 237 (90.8) | | |
| **Number and type of placements before adoption** | | | | | | 0.928 |
| Foster home | 114 (8.8) | 19 (16.7) | | 95 (83.3) | | |
| Orphanage | 717 (55.6) | 116 (16.2) | | 601 (83.8) | | |
| Many placements | 459 (35.6) | 71 (15.5) | | 388 (84.5) | | |
| **Parents** | | | | | | |
| **Adoptive family's socioeconomic status** | | | | | | 0.605 |
| Upper middle class | 747 (59) | 122 (16.3) | | 625 (83.7) | | |
| Lower middle class | 261 (20.6) | 40 (15.3) | | 221 (84.7) | | |
| Working class | 224 (17.7) | 33 (14.7) | | 191 (85.3) | | |
| Other | 34 (2.7) | 8 (23.5) | | 26 (76.5) | | |
| **Adoptive parent's marital status** | | | | | | 0.133 |
| Married parents | 1100 (86.5) | 169 (15.4) | | 931 (84.6) | | |
| Single-parent household | 171 (13.5) | 34 (19.9) | | 137 (80.1) | | |

*Deafness or impaired hearing in both ears, blindness or visual defect in both eyes, mental retardation or intellectual disability, and autism.

years of life are also considered to be a considerably important time for the maturation of the gut microbiome and the immune system [43].

Intestinal parasitic infections may disturb intestinal absorption processes. Although the presence of anemia could not to explain the association between intestinal parasite infections and later behavioral problems (Table 2), it is possible that intestinal parasitic infections contribute to later behavioral symptoms through micronutrient deficiencies, such as iron deficiency without anemia [20]. Fuglestad and coauthors (2008) showed that iron deficiency among internationally adopted children was more difficult to remedy than among adopted children with intestinal *Giardia* parasitism at the time of arrival. Later, this group (2013) showed that children with iron deficiency at adoption or 6 months later were more likely to score below average on tests of learning capacity 6 to 7 months post-adoption, and only 20% of the children with iron deficiency were anemic. Iron deficiency also has an adverse impact on

**Table 2. Associations between parasite status and the CBCL symptoms among international adoptees (n = 1302–1069).** Unstandardized regression coefficients (b), 95% confidence intervals (95% CI), and p values are reported.

| | Step 1, n = 1302 | | | Step 2, n = 1285 | | | Step 3, n = 1259 | | | Step 4, n = 1069 | | |
|---|---|---|---|---|---|---|---|---|---|---|---|---|
| | Externalizing problems b (95%CI) p | Internalizing problems b (95%CI) p | CBCL total score b (95% CI) p | Externalizing problems b (95%CI) p | Internalizing problems b (95%CI) p | CBCL total score b (95% CI) p | Externalizing problems b (95%CI) p | Internalizing problems b (95%CI) p | CBCL total score b (95% CI) p | Externalizing problems b (95%CI) p | Internalizing problems b (95%CI) p | CBCL total score b (95% CI) p |
| **Parasite infection on arrival** | 2.19 (0.81 to 3.56) 0.002 | 1.28 (0.43 to 2.13) 0.003 | 5.41 (2.09 to 8.73) 0.001 | 2.33 (0.99 to 3.67) 0.001 | 1.55 (0.69 to 2.41) <0.001 | 6.31 (3.19 to 9.50) <0.001 | 2.22 (0.86 to 3.57) <0.001 | 1.50 (0.63 to 2.36) 0.001 | 6.19 (3.00 to 9.39) <0.001 | 2.11 (0.72 to 3.49) 0.003 | 1.02 (0.13 to 1.90) 0.024 | 5.36 (2.06 to 8.66) 0.001 |
| **Age at the time of the questionnaire study was conducted** | | | | -0.25 (-0.38 to -0.13) <0.001 | 0.00 (-0.08 to 0.08) 0.989 | -0.27 (-0.57 to 0.03) 0.074 | -0.25 (-0.38 to -0.12) <0.001 | 0.00 (-0.08 to 0.08) 0.999 | -0.26 (-0.57 to 0.04) 0.086 | -0.38 (-0.52 to -0.24) <0.001 | -0.09 (-0.17 to 0.00) 0.062 | -0.59 (-0.92 to -0.25) 0.001 |
| **Age on arrival in Finland** | | | | 0.57 (0.24 to 0.89) 0.001 | 0.04 (-0.17 to 0.25) 0.685 | 1.07 (0.31 to 1.84) 0.006 | 0.54 (0.21 to 0.87) 0.001 | 0.05 (-0.16 to 0.26) 0.645 | 1.04 (0.26 to 1.81) 0.009 | 0.82 (0.45 to 1.19) <0.001 | 0.20 (-0.03 to 0.44) 0.089 | 1.69 (0.81 to 2.56) <0.001 |
| **Gender** | | | | 2.67 (1.67 to 3.67) <0.001 | -0.31 (-0.95 to 0.33) 0.350 | 4.56 (2.20 to 6.91) <0.001 | 2.70 (1.68 to 3.71) <0.001 | -0.30 (-0.95 to 0.36) 0.372 | 4.61 (2.22 to 7.01) <0.001 | 1.92 (0.85 to 2.99) <0.001 | -0.45 (-1.13 to 0.23) 0.197 | 3.09 (0.55 to 5.64) 0.017 |
| **Continent of birth (Africa)**[a] | | | | 0.69 (-0.95 to 2.33) 0.409 | -0.23 (-1.28 to 0.82) 0.662 | 0.46 (-3.40 to 4.31) 0.817 | 0.75 (-0.94 to 2.44) 0.385 | -0.44 (-1.52 to 0.64) 0.425 | 0.07 (-3.92 to 4.05) 0.974 | 1.29 (-0.51 to 3.09) 0.159 | -0.13 (-1.28 to 1.02) 0.827 | 1.78 (-2.51 to 6.07) 0.416 |
| **Continent of birth (South America)**[a] | | | | 2.29 (0.79 to 3.79) 0.003 | 0.05 (-0.91 to 1.01) 0.913 | 3.55 (0.02 to 7.08) 0.049 | 2.27 (0.75 to 3.79) 0.003 | 0.11 (-0.86 to 1.09) 0.822 | 3.70 (0.12 to 7.29) 0.043 | 2.10 (0.52 to 3.69) 0.009 | -0.02 (-1.03 to 0.99) 0.967 | 3.02 (-0.76 to 6.80) 0.117 |
| **Continent of birth (Eastern Europe)**[a] | | | | 2.45 (0.98 to 3.93) 0.001 | 0.66 (-0.28 to 1.60) 0.171 | 6.93 (3.47 to 10.39) <0.001 | 2.46 (0.96 to 3.95) 0.001 | 0.59 (-0.37 to 1.54) 0.229 | 6.91 (3.39 to 10.43) <0.001 | 2.76 (1.17 to 4.36) 0.001 | 0.64 (-0.38 to 1.66) 0.220 | 7.12 (3.31 to 10.92) <0.001 |
| **Pre-adoption placements (Orphanage)**[b] | | | | 0.37 (-1.39 to 2.13) 0.689 | -0.13 (-1.26 to 0.99) 0.817 | 0.76 (-3.38 to 4.91) 0.718 | 0.22 (-1.57 to 2.01) 0.809 | -0.15 (-1.30 to 1.00) 0.799 | 0.43 (-3.79 to 4.65) 0.841 | 0.01 (-1.83 to 1.85) 0.989 | -0.00 (-1.18 to 1.17) 0.998 | 0.17 (-4.21 to 4.56) 0.938 |
| **Pre-adoption placements (Many)**[b] | | | | 0.48 (-1.35 to 2.30) 0.608 | 0.37 (-0.79 to 1.54) 0.530 | 1.51 (-2.78 to 5.80) 0.489 | 0.31 (-1.54 to 2.16) 0.743 | 0.33 (-0.85 to 1.52) 0.581 | 1.08 (-3.28 to 5.44) 0.628 | -0.32 (-2.23 to 1.59) 0.741 | 0.23 (-0.99 to 1.45) 0.710 | 0.00 (-4.55 to 4.55) 0.999 |
| **Disabilities** | | | | 6.72 (5.09 to 8.35) <0.001 | 4.27 (3.23 to 5.32) <0.001 | 22.99 (19.15 to 26.84) <0.001 | 6.50 (4.85 to 8.16) <0.001 | 4.20 (3.14 to 5.26) <0.001 | 22.49 (18.60 to 26.38) <0.001 | 5.96 (4.18 to 7.75) <0.001 | 3.22 (2.08 to 4.36) <0.001 | 20.99 (16.73 to 25.25) <0.001 |

*(Continued)*

Table 2. (Continued)

| | Step 1, n = 1302 | | | Step 2, n = 1285 | | | Step 3, n = 1259 | | | Step 4, n = 1069 | | |
|---|---|---|---|---|---|---|---|---|---|---|---|---|
| | Externalizing problems b (95%CI) p | Internalizing problems b (95%CI) p | CBCL total score b (95% CI) p | Externalizing problems b (95%CI) p | Internalizing problems b (95%CI) p | CBCL total score b (95% CI) p | Externalizing problems b (95%CI) p | Internalizing problems b (95%CI) p | CBCL total score b (95% CI) p | Externalizing problems b (95%CI) p | Internalizing problems b (95%CI) p | CBCL total score b (95% CI) p |
| **Adoptive family's socioeconomic status** | | | | | | | 0.13 (-0.43 to 0.69) 0.653 | 0.17 (-0.19 to 0.53) 0.343 | 0.62 (-0.70 to 1.95) 0.355 | 0.25 (-0.32 to 0.82) 0.390 | 0.27 (-0.09 to 0.64) 0.144 | 0.93 (-0.43 to 2.29) 0.181 |
| **Adoptive parent's marital status** | | | | | | | -0.08 (-1.54 to 1.39) 0.919 | -1.21 (-2.15 to -0.28) 0.011 | -2.70 (-6.14 to 0.74) 0.124 | -0.15 (-1.72 to 1.42) 0.851 | -1.48 (-2.48 to -0.48) 0.004 | -3.09 (-6.82 to 0.65) 0.105 |
| **Anemia of child** | | | | | | | | | | -0.58 (-3.34 to 2.18) 0.679 | -0.34 (-2.11 to 1.42) 0.702 | -1.86 (-8.43 to 4.70) 0.578 |
| **Child's growth in terms of weight** | | | | | | | | | | -0.16 (-0.61 to 0.29) 0.481 | -0.35 (-0.64 to -0.06) 0.018 | -0.75 (-1.82 to 0.33) 0.172 |
| **$R^2/R^2$ adjusted** | 0.01 | 0.01 | 0.01 | 0.13 | 0.07 | 0.18 | 0.13 | 0.08 | 0.18 | 0.13 | 0.08 | 0.18 |

[a]reference: Asia,

[b]reference: Foster home.

neurobehavioral predictability for months after adoption and exhibits long-lasting effects on children's development [21–23]. Delays in cognitive development and problems in executive functions have repeatedly been shown to predict behavioral problems [44,45]. Thus, delays in cognitive development may also represent a physiological mechanism linking the nutrient deficiencies caused by intestinal parasites to later behavioral problems also in the adopted children.

As mentioned previously, the other plausible mechanism behind the association between intestinal parasite infection and later behavioral symptoms involves changes in intestinal microbiota caused by intestinal parasite infections. Intestinal parasites are shown to cause imbalance in microbiota and alterations to the intestinal microbiome by modulating the intestinal microbiota [27,29,30] and inducing intestinal inflammation [42]. Although research in this area is still limited, some reports indicate that gut microbiota and intestinal inflammation may be associated with neuropsychiatric and neurodevelopmental conditions [31,32] and even adulthood personality traits [33]. Further, microbial dysbiosis in adults has been associated with later anxiety [34] or depression-related behaviors [35]. In their pre-adoptive surroundings, the internationally adopted children have been already exposed early in life to alterations caused by intestinal parasites on their intestinal microbiome, and this early exposure also impacts the developing brain through gut-brain axis.

The use of antibiotics is also an alternative explanatory mechanism between the association with intestinal parasitic infections and later behavioral problems in our study population. Intestinal parasitic infections are often treated with antibiotics, and the use of antibiotics in the early years of life is known to have a long-lasting influence on gut microbiota, the intestinal microbiome, and overall health [41,42]. Recent studies have also shown an association between antibiotic treatment in infancy and later neurodevelopmental difficulties, such as behavioral problems [46,47].

The prevalence of intestinal parasites among internationally adopted children on arrival in Finland was 15.6%, which is comparable to the prevalence in adopted children in earlier studies in the other countries [3,9–11,48] but lower than the reported in some recent works [6,12–15]. This difference may be attributed to the development of better stool screening technologies, such as DNA-based methods, or reporting biases. The continent of origin of the adopted children was associated with the prevalence of parasites on arrival (Table 1). Similar to already reported data [3,4,6,9–15,48], the most prevalent pathogenic intestinal parasite detected among the adopted children was *Giardia lamblia*, but the parents were unable to specify the species of the parasites or worms in 35% of the cases. Not all intestinal parasites have the same pathology or effect on their hosts. However, the parasites found among these children are globally regarded as pathogens that enhance imbalances in the gut microbiota composition and dysfunction [11,49]. Although it remains uncertain whether dysbiosis is a cause or a consequence of intestinal inflammation, there are indications about its association with a disruption of the gut barrier. For their part, gut barrier disruptions may even contribute to the development of brain disorders or psychiatric symptoms [31,37]. Still, it is possible that the effects of different parasites are not similar. The association of the CBCL scores and individual parasite type is only suggestive, as many parents could not name the parasite type. Our finding may, however, support further developments pertaining to the gut-brain theory.

Public health care is free for all residents in Finland, and the first medical examinations of adopted children usually occur in public health centers supervised by medical professionals according to the recommendations of the Finnish adoption organizations. These guidelines are comparable with the screening protocols of newly arrived adopted children internationally [50]. Thus, it was assumed that any intestinal parasites found during the medical screenings were adequately treated. However, this study could not record such actual treatments.

Moreover, we are unaware if any children had been treated for the parasite infections in their country of origin prior to travelling to Finland. Little is also known about the pre-adoptive conditions of these children overall, but the information regarding early placement in an orphanage or a foster home was included in the models. The older the child at adoption, the longer the time spent in an orphanage or a foster home, and consequently, the longer the exposure time to intestinal parasites. Therefore, the age of the child on arrival in Finland was used as a measure of the exposure time in the analysis. The stronger CBCL association observed in the older children may be due to their longer exposures to parasite infections.

In this study, the CBCL questionnaire was considered an appropriate indicator of a child's later psychological symptoms, since it is a widely used, valid, and well-known method to evaluate children's behavioral and emotional problems, and is also considered suitable for questionnaire-based studies [8]. The grouping used here (internalizing, externalizing, and total score) is commonly applied in other research [7]. In a former meta-analysis, the behavior problems of internationally adopted children were measured using the CBCL, and the related measures showed only small differences between the outcomes of the adopted children and non-adopted, parent-reared children [7]. In this study, the behavior of the internationally adopted children was evaluated using the CBCL questionnaire at the time of evaluation, approximately 5 years after adoption. This finding also underscores the effect of intestinal parasites with regard to brain development in the early years regardless of the influence mechanisms of the intestinal parasites.

This study uncovered associations between an earlier intestinal parasite infection and higher CBCL problem scores in all three CBCL categories (externalizing problems, internalizing problems and CBCL total scores). The association between the CBCL symptoms and the parasite infection in the children adopted from South America or Eastern Europe may be due to the differences in the local parasites or the combination of the parasite infection with other risk factors, such as being exposed to drugs or alcohol in utero [4,51].

Other limitations of this study are also worth mentioning. This study was partly retrospective, as the health of the adopted children upon their arrival in Finland was ascertained from their parents' responses to questions, and the results were derived from parental-reported questionnaires; thus, this method is vulnerable to the inaccurate memories of the parents. The time delay from the arrival date to these responses (approximately 5 years) may also have resulted in a bias. However, we have found that the most of the adoptive parents provided exact details from their notes, documented laboratory results, or memories of the adopted child. The parents answered the questions precisely and had good knowledge of the health and medical issues of their adopted children because they had undergone substantial obligatory training before the adoption. Nonetheless, detailed medical nomenclature that does not hold significance for the parents may be lost. Almost 30% of the parents could not name the parasite found. However, intestinal parasites are extremely rare in Finland in general, and therefore, it is very likely that the parents of these children remembered whether their child had an intestinal parasite infection after arrival and the child was treated. We also tried to ascertain whether the studied children were anemic or had iron deficiency on arrival but doing so was not possible as their iron status was not registered.

We conducted a questionnaire study that queried many adoption-specific questions, which did not allow us to create a control group of non-adopted children with or without parasitic infections. Notably, no historic comparison groups were available in Finland or neighbouring countries, as intestinal parasites among children in Nordic countries were extremely rare at the time of adoption. The only foreign children (those of refugees or asylum seekers) were very few in number at the time of this study's enrolment. Thus, this work is the first to report on the association between intestinal parasite infections and subsequent behavioral problems among children.

This study uncovered associations between an earlier intestinal parasite infection and higher CBCL problem scores in all three CBCL categories (externalizing problems, internalizing problems, and CBCL total scores). The explained variance of the behavioral problems was relatively low in our regression models. Behavioral problems are caused by multiple factors, including the child's individual, parenting, and social factors [52]; however, we could not consider these aspects in our analyses.

Earlier international studies using standardized measures such as the CBCL to evaluate children coming from different continents suggest that no significant differences exist in the extent or types of symptoms [53]. Thus, the differences in"normality" in behavior between various continents are rather small and difficult to evaluate. Similar to other studies, we used the CBCL as a continuous measure for the two border groups and the total score [7]. However, we were unable to determine specific cut-off points for clinically relevant behavioral problems from these scores. However, such measures have been shown to predict clinical outcomes.

## Conclusion

Our results of a five-year follow up post-adoption suggest that intestinal parasite infections among internationally adopted children are associated with higher behavioral problem scores. The association persisted even after these children moved to a country with a low prevalence of parasites. The effect sizes of intestinal parasite infections on behavioral problems were relatively small. Behavioral problems are obviously multifactorial, and the finding that intestinal parasite infections can be strong predictors of behavioral problems at some point later in life was unexpected. However, the association between intestinal parasites and later behavioral problems was stronger than that between intestinal parasites and any other factors measured in this study, except disability. This suggests that internationally adopted children should be screened for intestinal parasites, and intestinal parasitism should be appropriately treated soon after arrival, especially among these at-risk populations, as the children are typically undergoing rapid growth and brain development at that age.

## Supporting information

**S1 File.**
(R)

## Acknowledgments

The authors are thankful to the all the internationally adopted children and their parents for the participation in this study.

## Author Contributions

**Conceptualization:** Marko Elovainio, Helena Lapinleimu.

**Data curation:** Marko Elovainio, Jaakko Matomäki, Helena Lapinleimu.

**Formal analysis:** Anna-Riitta Heikkilä, Marko Elovainio, Jaakko Matomäki.

**Funding acquisition:** Anna-Riitta Heikkilä, Jari Sinkkonen, Helena Lapinleimu.

**Investigation:** Anna-Riitta Heikkilä, Marko Elovainio, Hanna Raaska, Jari Sinkkonen, Helena Lapinleimu.

**Methodology:** Anna-Riitta Heikkilä, Marko Elovainio, Hanna Raaska, Helena Lapinleimu.

**Project administration:** Helena Lapinleimu.

**Resources:** Marko Elovainio, Helena Lapinleimu.

**Software:** Marko Elovainio.

**Supervision:** Marko Elovainio, Helena Lapinleimu.

**Validation:** Marko Elovainio, Hanna Raaska, Helena Lapinleimu.

**Visualization:** Anna-Riitta Heikkilä.

**Writing – original draft:** Anna-Riitta Heikkilä, Marko Elovainio, Helena Lapinleimu.

**Writing – review & editing:** Anna-Riitta Heikkilä, Marko Elovainio, Hanna Raaska, Jaakko Matomäki, Jari Sinkkonen, Helena Lapinleimu.

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
