## [Decision Letter · Decision Letter 0]

10 Aug 2020

PONE-D-20-08573

Intestinal parasites associated with later behavioral problems in internationally adopted children

PLOS ONE

Dear Dr. Heikkilä,

Thank you for submitting your manuscript to PLOS ONE. After careful consideration, we feel that it has merit but does not fully meet PLOS ONE’s publication criteria as it currently stands. Therefore, we invite you to submit a revised version of the manuscript that addresses the points raised during the review process.

Both reviewers appreciated the importance of the topic addressed and the way that the manuscript was written. They also both pointed out issues with the statistical analysis used that need to be addressed. In addition to the reviewers comments, I had several concerns about the study presented. Some important aspects have been left out of the current draft of the manuscript such as a discussion of limitations. This should be added and should address weaknesses such as the fact that the results are based on a parent-reported questionnaire. I was also concerned that some important confounders are not discussed; not all parasite infections have the same pathology or ultimate effect on the host, time spent in an orphanage or foster home appear not to have been considered, and the environment prior to adoption is not discussed. Were these confounders considered in the analysis and interpretation of the study? Lastly, a control was not used, such as a group of children not adopted but infected with parasites. Is there a reason a control was not used?

We look forward to receiving your revised manuscript.

Kind regards,

Adler R. Dillman

Academic Editor

PLOS ONE

Additional Editor Comments:

There are some comments that you will address in the body of the manuscript especially the Data analyses section, and the editors comments above regarding a discussion of limitations, confounding variables, and lack of a control.

2. Please provide additional details regarding participant consent.

In the ethics statement in the Methods and online submission information, please ensure that you have specified whether consent was informed.

3. PLOS ONE's publication criteria require that experiments, statistics, and other analyses are performed to a high technical standard; sample sizes are large enough to produce robust results; and methods are described in sufficient detail to allow another researcher to reproduce the experiment (http://journals.plos.org/plosone/s/criteria-for-publication#loc-3).

Moreover, the data presented in PLOS ONE manuscripts must support the conclusions drawn (http://journals.plos.org/plosone/s/criteria-for-publication#loc-4).

To that effect, please further discuss possible limitations to your study. In particular, the results are based on parent-reported questionnaires, without further consideration of time in an orphanage or foster home prior to adoption; the effect of different parasites (e.g., Giardia v. tape worm) and the time they affected their hosts are not distinguished. Thank you for your attention to this request.

Reviewers' comments:

Reviewer's Responses to Questions

**Comments to the Author**

1. Is the manuscript technically sound, and do the data support the conclusions?

Reviewer #1: Partly

2. Has the statistical analysis been performed appropriately and rigorously? 

Reviewer #1: No

3. Have the authors made all data underlying the findings in their manuscript fully available?

Reviewer #1: Yes

4. Is the manuscript presented in an intelligible fashion and written in standard English?

Reviewer #1: Yes

5. Review Comments to the Author

Reviewer #1: This is a well written manuscript and the participation rate for the study appears reasonable. The major analysis tool is multiple regression with the results seen in Table 2. There are several major issues. Specifically:

1. The regression sequence is well structured. However, the largest R-square is seen as 0.18, which means that quite a bit of the variation of the dependent factor is not being explained by this data. The authors give little or no attention to the R-square and should do so to explain other factors not included in the analysis which may be influencing the results.

2. The investigators mention a sensitivity analysis on page 7. They note that they assess the interaction effect between factors related to the child’s background and

intestinal parasite infection on the scales provided by the CBCL. The only interaction they mention is on page 10, lines 191-194. Exactly where is the sensitivity analysis and how is it explained in the ‘Results’ section?

3. There is little or no attention given to explaining the limitations of this study in the ‘Discussion’ section of the manuscript.

6. PLOS authors have the option to publish the peer review history of their article (what does this mean?). If published, this will include your full peer review and any attached files.

Reviewer #1: No

---

## [Author Response · Author response to Decision Letter 0]

4 Nov 2020

Here are our responses to the academic editor's and reviewers' comments. All the our point-by-point responses has been also attached as a separate file labeled as response to reviewers.

Some important aspects have been left out of the current draft of the manuscript such as 

(1) a discussion of limitations. This should be added and should address weaknesses such as the fact that the results are based on a parent-reported questionnaire. 

Our response: We thank the Editor and Reviewer 1 for raising this important question. We have now clarified and widened the discussion, especially with respect to the limitations, as follows (page 17, lines 320-358): 

“Other limitations of this study are also worth mentioning. This study was partly retrospective, as the health of the adopted children upon their arrival in Finland was ascertained from their parents’ responses to questions, and the results were derived from parental-reported questionnaires: thus, this method is vulnerable to the inaccurate memories of the parents. The time delay from the arrival date to these responses (approximately 5 years) may also have resulted in a bias. However, we have found that most of the adoptive parents provided exact details from their notes, documented laboratory results, or memories of the adopted child. The parents answered the questions precisely and had good knowledge of the health and medical issues of their adopted children because they had undergone substantial obligatory training before the adoption. Nonetheless, detailed medical nomenclature that does not hold significance for the parents may be lost. Almost 30 % of the parents could not name the parasite found. However, intestinal parasites are extremely rare in Finland in general, and therefore, it is very likely that the parents of these children remembered whether their child had an intestinal parasite infection after arrival and the child was treated. We also tried to ascertain whether the studied children were anemic or had iron deficiency on arrival but doing so was not possible as their iron status was not registered. 

We conducted a questionnaire study that queried many adoption-spesific questions, which did not allow us to create a control group of non-adopted children with or without intestinal parasite infection. Notably, no historic comparison groups were available in Finland or in neighbouring countries, as intestinal parasites among children in Nordic countries were extremely rare at the time of adoption. The only foreign children (those of refugees or asylum seekers) were very few in number at the time of this study’s enrolment. Thus, this work is the first to report on the association between intestinal parasite infections and subsequent behavioral problems among children.

This study uncovered associations between an earlier intestinal parasite infection and higher CBCL problem scores in all three CBCL categories (externalizing problems, internalizing problems, and CBCL total scores). The explained variance of behavioral problems was relatively low in our regression models. Behavioral problems are caused by multiple factors, including the child’s individual, parenting, and social factors [52]; however, we could not consider these aspects in our analyses.

Earlier international studies using standardized measures such as the CBCL to evaluate children coming from different continents suggest that no significant differences exist in the extent or types of symptoms [53]. Thus, the differences in ”normality” in behavior between various continents are rather small and difficult to evaluate. Similar to other studies, we used the CBCL as a continuous measure for the two border groups and the total score [7]. However, we were unable to determine specific cut-off points for clinically relevant behavioral problems from these scores. However, such measures have been shown to predict clinical outcomes.” 

(2) I was also concerned that some important confounders are not discussed; (a) not all parasite infections have the same pathology or ultimate effect on the host. 

a) Our response: These are very important points. We have now added to Discussion (page 15, lines 276-285): 

“Not all intestinal parasites have the same pathology or effect on their hosts. However, the parasites found among these children are globally regarded as pathogens that enhance imbalances in the gut microbiota composition and dysfunction [11, 49]. Although it remains uncertain whether dysbiosis is a cause or a consequence of intestinal inflammation, there are indications about its association with a disruption of the gut barrier. For their part, gut barrier disruptions may even contribute to the development of brain disorders or psychiatric symptoms [31, 37]. Still, it is possible that the effects of different parasites are not similar. The association of the CBCL-score and individual parasite type is only suggestive as many parents could not name the parasite type. Our finding may, however, support further developments pertaining to the gut-brain theory.”

In addition to assess whether different parasites affected the CBCL scores differently for these internationally adopted children, we divided the detected parasites into protozoans and helminths, and tested the largest individual groups, namely Giardia and helminths. 

Grouping of intestinal parasites detected from the adopted children into protozoans and helminths.

Parasite species n (%) Grouping

Giardia lamblia 122 (59.2) protozoan

Entamoeba histolytica 13 (6.3) protozoan

Blastocystis hominis 5 (3.4) protozoan

 Hymenolepis nana 2 (1.0) helminth

Strongyloides 4 (1.9) helminth

Tapeworm 5 (2.4) helminth

Enterobius vermicularis 2 (2) helminth

Ascarias lumbricoides 1 (0.5) helminth

Schistosomiasis 1 (0.5) helminth

Whipworm 1 (0.5) helminth

Unidentified worm (anonymous) 17 (8.3) helminth

Anonymous parasite 62 (30.1) protozoan

As shown below, no significant differences were observed in the CBCL categories between the children with Giardia infections (n = 118) and those with helminth infections (n = 34). Children affected by multiple parasites were excluded from this analysis.

Tabulated differences in the CBCL scores between internationally adopted children with detected with Giardia infections (n = 118) and helminth infections (n = 34). Children with both parasitic infections were excluded from this analysis. We report standard errors (SEs), t-values, p-values, and 95% confidence intervals (95%CIs).

CBCL Scores Estimate SE t-value p-value 95%CI

CBCL Total helminth vs giardia -0.559 4.739 -0.118 0.906 -9.930 to 8.812

External helminth vs giardia -1.232 1.830 -0.674 0.502 -4.851 to 2.386

Internal helminth vs giardia 0.208 1.279 0.163 0.871 -2.320 to 2.736

This information has been added in Results as follows: “The associations found between the CBCL categories and parasites did not to differ significantly between the parasite types, namely Giardia and helminth” (page 10, lines 193-195).

(b) time spent in an orphanage or foster home appear not to have been considered, and the environment prior to adoption is not discussed. Were these confounders considered in the analysis and interpretation of the study?

(b) Our response: We have now added the following text to Methods (page 7, lines 160-165):

“The older the child at adoption, the longer the time spent in an orphanage or a foster home and the longer the exposure time to intestinal parasites. Therefore, the age of the child on arrival in Finland was used as a measure of exposure time in the analysis. Little is known about the pre-adoptive conditions of the children, but the early placement in an orphanage or foster home was considered in the models.”

The following text was also added to Discussion (page 16, lines 294-300): “Little is also known about the pre-adoptive conditions of these children, but the information regarding early placements in an orphanage or a foster home was included in the models. The older the child at adoption, the longer the time spent in an orphanage or a foster home, and consequently, the longer the exposure time to intestinal parasites. Therefore, the age of the child on arrival in Finland was used as a measure of the exposure time in the analysis. The stronger CBCL association observed in the older children may be due to their longer exposures to parasite infections.”

(3) Lastly, a control was not used, such as a group of children not adopted but infected with parasites. Is there a reason a control was not used?

Our response: We thank the Editor of raising up this question. We have now added the following text to Discussion (page 17, lines 337-344): “We conducted a questionnaire study that queried many adoption-specific questions, which did not allow us to create a control group of non-adopted children with or without parasitic infections. Notably, no historic comparison groups were available in Finland or neighbouring countries, as intestinal parasites among children in Nordic countries were extremely rare at the time of adoption. The only foreign children (those of refugees or asylum seekers) were very few in number at the time of this study’s enrolment. Thus, this work is the first to report on the association between intestinal parasite infections and subsequent behavioral problems among children.”

(4) Our response: There is one change to the financial disclosure and the conflict of interest: Helena Lapinleimu (MD, PhD) has received consultation fees from the following national adoption organizations in Finland: Interpedia, Save the Children Association Finland., and the international adoption service of the City of Helsinki. 

(5) If applicable, we recommend that you deposit your laboratory protocols in protocols.io to enhance the reproducibility of your results. Protocols.io assigns your protocol its own identifier (DOI) so that it can be cited independently in the future. 

Our response: The answers of the parents are based on general good laboratory practice in Finland. Only a handful of central laboratories in Finland perform all the analyses pertaining to parasite samples, and all of them use very similar clinical protocols. Parasite ova samples were sent to the laboratories following the Finnish general guidelines and were analyzed under a microscope to detect the parasite and its type.

Additional Editor comments: 

(6) There are some comments that you will address in the body of the manuscript especially the Data analyses section, and the editor’s comments above regarding a discussion of the limitations, confounding variables, and lack of a control. 

Our response: We have made the recommended changes in Data analyses and have modified the sections discussing the results and the limitations of the study. 

Our response: These requirements have been taken into account.

2. Please provide additional details regarding participant consent.

In the ethics statement in the Methods and online submission information, please ensure that you have specified whether consent was informed.

Our response: Methods in the manuscript states, that “The study was approved by the Ethics Review Committee of the Hospital District of Southwest Finland, and the participating children and their parents provided their written consent to participate.”

3. PLOS ONE's publication criteria require that experiments, statistics, and other analyses are performed to a high technical standard; sample sizes are large enough to produce robust results; and methods are described in sufficient detail to allow another researcher to reproduce the experiment (http://journals.plos.org/plosone/s/criteria-for-publication#loc-3). 

Our response: The analyses codes are now attached as supporting information.

Moreover, the data presented in PLOS ONE manuscripts must support the conclusions drawn (http://journals.plos.org/plosone/s/criteria-for-publication#loc-4). 

To that effect, please further discuss possible limitations to your study. In particular, the results are based on parent-reported questionnaires, without further consideration of time in an orphanage or foster home prior to adoption; the effect of different parasites (e.g., Giardia v. tape worm) and the time they affected their hosts are not distinguished. Thank you for your attention to this request.

Our response: These are valuable comments and have been taken into account in the revised manuscript. Please, see our previous responses in this document. 

Our response: We have provided the Data Availability statement below.

Data Availability

This manuscript is based on health data. Access to these data is regulated by Finnish legislation and Findata, the Health and Social Data Permit Authority. The disclosure of data to third parties without explicit permission from Findata is prohibited. Only those fulfilling the requirements established by Finnish legislation and Findata for viewing confidential data are able to access the data. See

https://www.findata.fi/en/about-us/data-protection-and-the-processing-of-personal-data/

Reviewer 1: This is a well written manuscript and the participation rate for the study appears reasonable. The major analysis tool is multiple regression with the results seen in Table 2. There are several major issues. Specifically:

1. The regression sequence is well structured. However, the largest R-square is seen as 0.18, which means that quite a bit of the variation of the dependent factor is not been explained by this data. The authors give little or no attention to the R-square and should do so to explain other factors not included in the analysis which may be influencing the results. 

Our response: The reviewer has brought up an important issue. The explained variance of behavioral problems was relatively low in our regression models, but that was more or less expected. Behavioral problems are caused by multiple factors, including a child’s individual, parenting, and social factors [52], which we could not consider in our analyses. We have stated this aspect in Discussion. 

We have explained the following regarding the limitations (page 18, lines 347-350): “The explained variance of the behavioral problems was relatively low in our regression models. Behavioral problems are caused by multiple factors, including the child’s individual, parenting, and social factors [52]; however, we could not consider these aspects in our analyses.”

2. The investigators mention a sensitivity analysis on page 7. The note that they assess the interaction effect between factors related to the child’s background and intestinal parasite infection on the scales provided by the CBCL. The only interaction they mention is on page 10, lines 191-194. Exactly where is the sensitivity analysis and how is it explained in the ’Results’ section. 

Our response: We thank the reviewer for this valuable comment. We have clarified the reporting of the other interaction effects in Methods (page 7, lines 157-160) 

“For the sensitivity evaluation, the interacting effects between the child’s intestinal parasite infection, age, gender, age at adoption, continent of birth, parental socioeconomic status, and parental marital status on the scales provided by the CBCL were assessed.” 

The section presenting the results now also explains the following (page 10, lines 198-210): 

“Significant interaction effects were observed between continents of birth and parasite infection for all CBCL categories: intestinal parasite infections increased external CBCL scores to a greater extent for children from South America (p = 0.010) than for children from Asia. For internalizing symptoms, intestinal parasite infections increased CBCL scores to a greater extent among children from South America (p = 0.016) and Eastern Europe (p = 0.043) than among those from Asia. The total CBCL symptoms scores were significantly higher (p = 0.003) among the parasite-infected adopted children from South America. The means of the externalizing symptoms, internalizing symptoms, and total CBCL symptoms were lower in parasite-infected children adopted from Asia than in those with a parasitic infection adopted from South America (9.6 vs 17.2, 6.2 vs 9.4, and 26.5 vs 45.3, respectively). Moreover, the mean of the internalizing symptoms (6.2) was lower in parasite-infected children adopted from Asia than in children with parasitic infection adopted from Eastern Europe (10.1).”

3. There is little or no attention given to explaining the limitations of this study in the ’Discussion’ section of the manuscript. 

Our response: We thank both, the Editor and Reviewer 1 for this important comment. We have addressed this shortcoming in the revised paper; please the material added to Discussion (beginning from page 15).

Reviewer 2: I read with the great interesting this original paper, which covers new point of views in the area of adoption and parasitic infections. I really love this paper and strongly suggest publication after a few revisions.

1. Major revision: the authors should perform a multivariate logistic regression analyses in order or understand how much the country of birth (or others factors) really contribute to the different scores.

Our response: We thank Reviewer 2 for raising an interesting question. We have conducted multivariate linear regression analyses, because all the outcomes are continuous variables. We have used categorical variables where appropriate and made changes to Table 2 to clarify the strong associations there are between the continent of birth and CBCL dimensions. We also changed the description about the interaction in Results, as follows (page 10, lines 198-210):

“Significant interaction effects were observed between continents of birth and parasite infection for all CBCL categories: intestinal parasite infections increased external CBCL scores to a greater extent for children from South America (p = 0.010) than for children from Asia. For internalizing symptoms, intestinal parasite infections increased CBCL scores to a greater extent among children from South America (p = 0.016) and Eastern Europe (p = 0.043) than among those from Asia. The total CBCL symptoms scores were significantly higher (p = 0.003) among the parasite-infected adopted children from South America. The means of the externalizing symptoms, internalizing symptoms, and total CBCL symptoms were lower in parasite-infected children adopted from Asia than in those with a parasitic infection adopted from South America (9.6 vs 17.2, 6.2 vs 9.4, and 26.5 vs 45.3, respectively). Moreover, the mean of the internalizing symptoms (6.2) was lower in parasite-infected children adopted from Asia than in children with parasitic infection adopted from Eastern Europe (10.1).”

All the other multivariate associations are reported in Table 2. To conduct a logistic regression analysis, all the outcomes would need to be converted into dichotomous variables. We are more than happy to additionally perform logistic regression analyses if the reviewer feels strongly about it, but we don’t quite understand how to justify this, as converting continuous variables to dichotomous ones would cause loss of information. 

2. Minor revisions: Please add the 95% CI to those results with significant P values (< 0.05).

Our response: We have now added all the 95% CI values to the Table 2.

---

## [Editor Report · Decision Letter 1]

2 Dec 2020

PONE-D-20-08573R1

Intestinal parasites associated with later behavioral problems in internationally adopted children

PLOS ONE

Dear Dr. Heikkilä,

Thank you for submitting your manuscript to PLOS ONE. After careful consideration, we feel that it has merit and has been significantly improved, but does not fully meet PLOS ONE’s publication criteria as it currently stands. Therefore, we invite you to submit a revised version of the manuscript that addresses the points raised during the review process.

In particular, while the revised version now includes a discussion of limitations and includes some caveats, I am concerned that the title and abstract might be over-interpreted. The limitations should be acknowledged in the abstract, and the title should be adjusted accordingly, to be less assertive title regarding the observed association discussed in the paper. The authors can decide how to do this, but one suggestion would be; "Intestinal parasites may be associated with later behavioral problems in internationally adopted children.."

We look forward to receiving your revised manuscript.

Kind regards,

Adler R. Dillman, Ph.D.

Academic Editor

PLOS ONE

Additional Editor Comments (if provided):

Thank you for including a detailed response to all concerns raised regarding the previous version of the paper. The revised manuscript is much improved.

---

## [Author Response · Author response to Decision Letter 1]

23 Dec 2020

We thank the Editor the proposal to change the title of the paper and we have made it as suggested: “Intestinal parasites may be associated with later behavioral problems in internationally adopted children”.

The abstract has now also been modified and the main limitations of the study have also been recognized in the abstract. The change is made in the abstract as follows, beginning from the methods paragraph (page 2, lines 33-50):

“Methods: Data for this study were sourced from the Finnish Adoption Study (FinAdo) based on parental questionnaires for all internationally adopted children under 18 years (n = 1450) who arrived in Finland from 1985 to 2007. A total of 1293 families provided sufficient information on the adoptee’s background, parasitic status on arrival, and behavioral symptoms at the median time of 5 years after arrival (mean age = 7.8 years). Behavioral and emotional disorders were evaluated with the Child Behavior Checklist (CBCL). Statistical analyses were conducted using linear regression. 

Results: Of the 1293 families, parents of 206 adoptive children reported intestinal parasites in their adopted children on arrival. Parasite-infected children had subsequently higher CBCL problem scores than the children without parasites (p < 0.001). The association between intestinal parasites and later behavioral problems was stronger than that between intestinal parasites and any other factors measured in this study, except disability. 

Limitations: The control group was naturally provided by the adopted children without parasite infections, but we could not compare the adopted children to non-adopted children without a defined parasite infection. We were unable to specify the effects associated with a specific parasite type. It was not possible either to include multiple environmental factors that could have been associated with behavioral problems in the models, which indicated only modest explanatory values. 

Conclusions: In this study, intestinal parasite infections in early childhood may be associated with children’s later psychological wellbeing, even in children who move to a country with a low prevalence of parasites. Our findings may support further developments pertaining to the gut-brain theory.”

---

## [Editor Report · Decision Letter 2]

8 Jan 2021

Intestinal parasites may be associated with later behavioral problems in internationally adopted children

PONE-D-20-08573R2

Dear Dr. Heikkilä,

We’re pleased to inform you that your manuscript has been judged scientifically suitable for publication and will be formally accepted for publication once it meets all outstanding technical requirements.

Kind regards,

Adler R. Dillman, Ph.D.

Academic Editor

PLOS ONE

Additional Editor Comments (optional):

Thank you for addressing the previous concerns and suggestions.
---

## [Editor Report · Acceptance letter]

14 Jan 2021

PONE-D-20-08573R2 

Intestinal parasites may be associated with later behavioral problems in internationally adopted children 

Dear Dr. Heikkilä:

I'm pleased to inform you that your manuscript has been deemed suitable for publication in PLOS ONE. Congratulations! Your manuscript is now with our production department. 

Kind regards, 

on behalf of

Dr. Adler R. Dillman 

Academic Editor

PLOS ONE